# Design and Synthesis of (*Z*)-2-(Benzylamino)-5-benzylidenethiazol-4(5*H*)-one Derivatives as Tyrosinase Inhibitors and Their Anti-Melanogenic and Antioxidant Effects

**DOI:** 10.3390/molecules28020848

**Published:** 2023-01-14

**Authors:** Jieun Lee, Yu Jung Park, Hee Jin Jung, Sultan Ullah, Dahye Yoon, Yeongmu Jeong, Ga Young Kim, Min Kyung Kang, Dongwan Kang, Yujin Park, Pusoon Chun, Hae Young Chung, Hyung Ryong Moon

**Affiliations:** 1Laboratory of Medicinal Chemistry, Department of Manufacturing Pharmacy, College of Pharmacy, Pusan National University, Busan 46241, Republic of Korea; 2Department of Pharmacy, College of Pharmacy, Pusan National University, Busan 46241, Republic of Korea; 3Department of Molecular Medicine, UF Scripps Biomedical Research, West Palm Beach, FL 33458, USA; 4Department of Medicinal Chemistry, New Drug Development Center, Daegu-Gyeongbuk Medical Innovation Foundation, Daegu 41061, Republic of Korea; 5College of Pharmacy and Inje Institute of Pharmaceutical Sciences and Research, Inje University, Gimhae 50834, Gyeongnam, Republic of Korea

**Keywords:** tyrosinase, melanin, BABT, antioxidant activity, in silico, MITF, Lineweaver–Burk plots

## Abstract

In this study, (*Z*)-2-(benzylamino)-5-benzylidenethiazol-4(5*H*)-one (BABT) derivatives were designed as tyrosinase inhibitors based on the structure of MHY2081, using a simplified approach. Of the 14 BABT derivatives synthesized, two derivatives ((*Z*)-2-(benzylamino)-5-(3-hydroxy-4-methoxybenzylidene)thiazol-4(5*H*)-one [**7**] and (*Z*)-2-(benzylamino)-5-(2,4-dihydroxybenzylidene)thiazol-4(5*H*)-one [**8**]) showed more potent mushroom tyrosinase inhibitory activities than kojic acid, regardless of the substrate used; in particular, compound **8** was 106-fold more potent than kojic acid when l-tyrosine was used as the substrate. Analysis of Lineweaver–Burk plots for **7** and **8** indicated that they were competitive inhibitors, which was confirmed via in silico docking. In experiments using B16F10 cells, **8** exerted a greater ability to inhibit melanin production than kojic acid, and it inhibited cellular tyrosinase activity in a concentration-dependent manner, indicating that the anti-melanogenic effect of **8** is attributable to its ability to inhibit tyrosinase. In addition, **8** exhibited strong antioxidant activity to scavenge 2,2-diphenyl-1-picrylhydrazyl and 2,2′-azino-bis(3-ethylbenzothiazoline-6-sulfonic acid) radicals and peroxynitrite and inhibited the expression of melanogenesis-associated proteins (tyrosinase and microphthalmia-associated transcription factor). These results suggest that BABT derivative **8** is a promising candidate for the treatment of hyperpigmentation-related diseases, owing to its inhibition of melanogenesis-associated protein expression, direct tyrosinase inhibition, and antioxidant activity.

## 1. Introduction

Melanin is a macromolecular pigment found in bacteria and mammals [1] that affects their phenotypic appearance. Among the five basic types of melanin, eumelanin, which has a black-to-brown color, and pheomelanin, which has a yellow-to-red color, are the most common [2]. Melanogenesis is a complex process that produces melanin in the melanosomes of melanocytes present in the bottom layer of the epidermis. Melanogenesis is initiated by ultraviolet (UV) radiation and α-melanocyte-stimulating hormone (α-MSH), and exposure to UV radiation is the most important extrinsic factor affecting melanogenesis [3,4]. Melanin exerts the beneficial effect of protecting the skin from harmful UV radiation of the sun. However, abnormal hyperpigmentation in specific regions of the skin causes aesthetic problems and hyperpigmentation-related conditions, such as melasma, freckles, and melanoma, the most serious type of skin cancer. Melanin is biosynthesized from l-tyrosine, a natural amino acid, via several complex chemico-enzymatic processes using melanogenesis-associated enzymes, such as tyrosinase, tyrosinase-related protein (TRP)-1, and TRP-2 [5]. During melanogenesis, dopaquinone is produced from l-tyrosine via a two-step enzymatic reaction catalyzed by tyrosinase (conversion of l-tyrosine into l-dopa using the monophenolase activity of tyrosinase and conversion of l-dopa into dopaquinone using the diphenolase activity of tyrosinase) and serves as a key intermediate for eumelanin and pheomelanin production. In the absence of sulfur nucleophiles, such as glutathione and cysteine, dopaquinone is modified via the Michael addition of an intramolecular amino group [6] and auto-oxidized to dopachrome, which has been used as a biomarker for measuring the inhibitory activities of tyrosinase inhibitors due to its strong absorbance at 475 nm [7].

Tyrosinase is a metalloenzyme containing two copper atoms in its active site, each of which is coordinated to the imidazole rings of three histidine residues [8]. Tyrosinase is found at different locations, depending on the type of species. For example, mushroom tyrosinase is located in the cytoplasm in a soluble tetrameric form, and mammalian tyrosinases, including human tyrosinase, are anchored to the membrane of melanosomes in a glycosylated monomeric form [9]. As tyrosinase acts as a rate-determining enzyme during melanin biosynthesis [10], it is a crucial determinant in regulating melanogenesis. Therefore, tyrosinase is considered the most important target for the inhibition of melanogenesis, and numerous natural and synthetic compounds have been identified as tyrosinase inhibitors. However, due to safety concerns, only a few compounds have been approved for the clinical treatment of hyperpigmentation-related diseases.

Over the past decade, we have prepared several compounds with β-phenyl-α,β-unsaturated carbonyl (PUSC) templates in our laboratory, which have shown potent anti-tyrosinase activities in vitro and/or in vivo [11,12,13,14,15,16,17]. Our data revealed that the PUSC template plays a core role in tyrosinase inhibitory activity. As shown in Figure 1, pethidine, a synthetic opioid analgesic, is designed using the simplified approach of cleaving the fused rings of morphine, an opioid analgesic. Because MHY2081 with the PUSC template exhibits strong inhibitory activities against mushroom tyrosinase and murine cellular tyrosinase [18], we used MHY2081 as a parent substance for the development of novel tyrosinase inhibitors in this study (Figure 1). For simplification of the MHY2081 chemical structure, we cleaved the benzothiazole ring via removal of nitrogen and sulfur atoms to obtain a simplified compound, (*Z*)-2-(benzylamino)-5-benzylidenethiazol-4(5*H*)-one (BABT). Because the BABT derivatives still contained a PUSC template, and a pharmacophore mapping technique considering the features of MHY2081 provided positive results, we expected them to show anti-tyrosinase potency. Therefore, 14 BABT derivatives were synthesized from 2-(benzylamino)thiazol-4(5*H*)-one, a key intermediate, and assayed to determine their anti-tyrosinase activities against mushrooms and mice, mode of action, anti-melanogenic and antioxidant activities, and expression levels of proteins and genes associated with melanogenesis. In addition, in silico docking simulations with mushroom tyrosinase and the previously prepared human tyrosinase homology model were performed using the Schrödinger Suite.

## 2. Results and Discussion

### 2.1. Chemistry

The preparation scheme for BABT derivatives **1**–**14** bearing a PUSC template is described in Figure 1. Notably, 2-(benzylamino)thiazol-4(5*H*)-one (**16**) was a common key intermediate for BABT derivatives. Compound **16** was prepared from *N*-benzyl-2-chloroacetamide (**15**) via migratory cyclization. The reaction of benzylamine with 2-chloroacetyl chloride in acetic acid gave compound **15** in moderate yield, which in turn was converted to the key intermediate, 2-(benzylamino)thiazol-4(5*H*)-one (**16**), via *N*-benzyl-2-thiocyanatoacetamide (**16′**; Figure 2), by reacting with ammonium thiocyanate in ethanol under reflux. A plausible mechanism for this migratory cyclisation [19] is depicted in Figure 2. The reaction of **15** with ammonium thiocyanate in ethanol under reflux yielded **16′** (an initial product), after which the thiocyanate nitrogen of **16′** attacked the carbonyl of amide and moved the benzylamine anion to the carbon of thiocyanate to generate **16** under reflux conditions. The desired BABT derivatives **1**–**7** and **9**–**14** were synthesized (yield: 36–93%) via a condensation reaction between **16** and **13** substituted benzaldehydes in the presence of sodium acetate in acetic acid under reflux. Derivative (*Z*)-2-(benzylamino)-5-(2,4-dihydroxybenzylidene)thiazol-4(5*H*)-one (**8**) with a 2,4-dihydroxyphenyl moiety at the β-position of the PUSC scaffold was obtained (yield: 65%) from (*Z*)-2-(benzylamino)-5-(2,4-dimethoxybenzylidene)thiazol-4(5*H*)-one (**10**) bearing a 2,4-dimethoxyphenyl ring via *O*-demethylation using boron tribromide at room temperature. The C=C bond geometry of BABT derivatives generated via coupling reactions of **16** with benzaldehydes was confirmed by ^3^*J*_(C4-Hβ)_ values of C4 in ^1^H-coupled ^13^C nuclear magnetic resonance (NMR) spectroscopy. According to previous reports [20], when the configuration of the carbonyl group is the same as that of β-hydrogen, *J* values appear in the range of 3.6–7.0 Hz, but when the configuration is reversed, *J* values are typically >10 Hz. To confirm the C=C bond geometry of the BABT derivatives, the ^1^H-coupled ^13^C NMR spectrum of derivative **8** was measured (see Appendix A). The peak of C4 appeared as a doublet by the coupling of H_β_ at 180.8 ppm, and the ^3^*J*_(C4-Hβ)_ value was 5.6 Hz, implying that the derivative has (*Z*)-geometry. The (*Z*)-isomer is thermodynamically more stable than the corresponding (*E*)-isomer because of the steric factor between the carbonyl group and phenyl ring and an intramolecular hydrogen bond between the oxygen of the carbonyl group and the vinylic H_β_ [21,22]. Additionally, the peak of H_β_ in the ^1^H NMR spectrum appeared downfield (7.48–7.83 ppm) due to the anisotropic effect of the carbonyl group. These findings suggest that the BABT derivatives are (*Z*)-isomers.

### 2.2. Mushroom Tyrosinase Inhibition of BABT Derivatives

As mushroom tyrosinase is commercially available and relatively cheap, it is used as a tool for the identification of novel tyrosinase inhibitors. Here, we examined the tyrosinase inhibitory activities of the 14 BABT derivatives against mushroom tyrosinase in the presence of l-tyrosine or l-dopa as substrates. Kojic acid, a well-known tyrosinase inhibitor, was used as a positive reference control because it is widely used as a positive reference control for tyrosinase inhibition assays. Since dopaquinone produced during melanogenesis strongly absorbs 475 nm light, the degree of tyrosinase inhibition of the test compounds was determined by measuring the optical density at 475 nm. Half-maximal inhibitory concentration (IC_50_) values were obtained from the inhibition tests performed at 3–6 different concentrations of the test compounds, and the results are shown in Table 1.

When l-tyrosine was provided as a substrate for mushroom tyrosinase, BABT derivative, (*Z*)-2-(benzylamino)-5-(4-hydroxybenzylidene)thiazol-4(5*H*)-one (**1**), bearing a 4-hydroxyl group on the β-phenyl ring of the PUSC template, showed an IC_50_ value of 27.5 ± 2.93 µM, which was approximately the same as that of kojic acid (IC_50_ = 28.6 ± 3.56 µM). However, the change of the 4-hydroxyl group in **1** to the corresponding 4-methoxyl group (BABT derivative (*Z*)-2-(benzylamino)-5-(4-methoxybenzylidene)thiazol-4(5*H*)-one [**2**]) resulted in 4.6-fold lower tyrosinase inhibitory activity. As observed in previous studies [17,23], the BABT derivatives without any hydroxyl groups on the β-phenyl ring of the PUSC template ((*Z*)-2-(benzylamino)-5-(2,4-dimethoxybenzylidene)thiazol-4(5*H*)-one [**10**], (*Z*)-2-(benzylamino)-5-(3,4-dimethoxybenzylidene)thiazol-4(5*H*)-one [**11**], and (*Z*)-2-(benzylamino)-5-(3,4,5-trimethoxybenzylidene)thiazol-4(5*H*)-one [**14**]) exhibited little-to-no mushroom tyrosinase inhibitory activity. The introduction of two methoxyl ((*Z*)-2-(benzylamino)-5-(4-hydroxy-3,5-dimethoxybenzylidene)thiazol-4(5*H*)-one [**12**]) or two *t*-butyl ((*Z*)-2-(benzylamino)-5-(3,5-di-*tert*-butyl-4-hydroxybenzylidene)thiazol-4(5*H*)-one [**13**]) groups at positions 3 and 5 on the β-phenyl ring of derivative **1** also decreased its tyrosinase inhibitory activity (**12**: IC_50_ = 166.0 ± 8.04 µM; **13**: IC_50_ = 136.7 ± 10.69 µM). Similarly, the insertion of an alkoxyl group (methoxyl or ethoxyl) at position 3 on the β-phenyl ring of derivative **1** led to low tyrosinase inhibition (**5**: IC_50_ = 148.3 ± 1.12 µM; **6**: IC_50_ = 198.9 ± 8.57 µM). Exchanging the two substituents (3-OMe and 4-OH) on the β-phenyl ring in derivative (*Z*)-2-(benzylamino)-5-(4-hydroxy-3-methoxybenzylidene)thiazol-4(5*H*)-one (**5**) resulted in 14.8-fold stronger tyrosinase inhibitory activity ((*Z*)-2-(benzylamino)-5-(3-hydroxy-4-methoxybenzylidene)thiazol-4(5*H*)-one [**7**]: IC_50_ = 10.0 ± 0.90 µM). Notably, an additional hydroxyl group at position 3 on the β-phenyl ring of derivative **1** reduced its tyrosinase inhibitory activity, but an additional hydroxyl group at position 2 on the β-phenyl ring of derivative **1** drastically increased its tyrosinase inhibitory activity ((*Z*)-2-(benzylamino)-5-(2,4-dihydroxybenzylidene)thiazol-4(5*H*)-one [**8**]: IC_50_ = 0.27 ± 0.03 µM; (*Z*)-2-(benzylamino)-5-(3,4-dihydroxybenzylidene)thiazol-4(5*H*)-one [**9**]: IC_50_ > 300 µM). When the two phenolic hydroxyl substituents of each derivative (**8** and **9**) were transformed into fluorine substituents, their tyrosinase inhibitory activities were drastically reduced ((*Z*)-2-(benzylamino)-5-(2,4-difluorobenzylidene)thiazol-4(5*H*)-one [**3**]: IC_50_ > 300 µM) or very weak ((*Z*)-2-(benzylamino)-5-(3,4-difluorobenzylidene)thiazol-4(5*H*)-one [**4**]: IC_50_ = 288.7 ± 14.72 µM). A hydroxyl substituent can act as a hydrogen bond donor and acceptor, whereas a fluorine substituent can only act as a hydrogen bond acceptor. Chemical structural differences between derivatives **3** and **8** and their tyrosinase inhibitory activities indicate that substituents at positions 2 and 4 of the β-phenyl ring of the PUSC template play important roles in its tyrosinase inhibitory activity. In particular, their ability to act as hydrogen bond donors has a decisive and significant effect on tyrosinase inhibition.

When l-dopa was used as a substrate for mushroom tyrosinase, the tested compounds generally showed higher IC_50_ values than those of tested compounds measured in the presence of l-tyrosine. BABT derivative **1** with a 4-hydroxyl substituent had a 2-fold higher IC_50_ value (65.1 ± 8.29 µM) compared to the case where l-tyrosine was used. The IC_50_ value of kojic acid (IC_50_ = 20.1 ± 0.46 µM) in the presence of l-dopa was similar to that observed in the presence of l-tyrosine. The conversion of 4-OH on the β-phenyl ring of **1** into a methoxy substituent decreased the tyrosinase inhibitory activity by 3.7-fold (**2**: IC_50_ = 239.4 ± 2.38 µM), indicating that the role of the 4-substituent as a hydrogen donor is important in conferring tyrosinase inhibitory activity. As observed in the results where l-tyrosine was used as a substrate for mushroom tyrosinase, derivatives (**2**–**4**, **10**, **11**, and **14**) without any hydroxyl substituent on the β-phenyl ring of the PUSC template showed no or weak tyrosinase inhibitory activities. Additional alkoxyl (OMe or OEt) or bulky alkyl (*t*-butyl) groups at position 3 or positions 3 and 5 on the β-phenyl ring of derivative **1** greatly reduced its tyrosinase inhibitory activity (**5**, **12**, and **13**: IC_50_ > 300 µM; (*Z*)-2-(benzylamino)-5-(3-ethoxy-4-hydroxybenzylidene)thiazol-4(5*H*)-one [**6**]: IC_50_ = 248.1 ± 11.54 µM), and the exchange of 3-OMe and 4-OH substituents of derivative **5** greatly enhanced its tyrosinase inhibitory activity (**9**: IC_50_ = 10.3 ± 0.24 µM), which was 2-fold more potent than that of kojic acid. The strongest tyrosinase inhibition was found in derivative **8** with a 2,4-dihydroxyphenyl moiety, and its IC_50_ value was 1.04 ± 0.05 µM. However, this value was 4-fold higher than that obtained using l-tyrosine as the substrate. Contrary to **8** with two hydroxyl groups at positions 2 and 4, derivative **9** with two hydroxyl groups at positions 3 and 4 had an IC_50_ value >300 µM, indicating that the positions of the hydroxyl substituents on the β-phenyl ring of the PUSC template sensitively affect the tyrosinase inhibitory activity regardless of the type of substrate (l-dopa or l-tyrosine) used.

### 2.3. Inhibitory Mechanisms of Derivatives ***7*** and ***8*** against Mushroom Tyrosinase

As **7** with a 3-hydroxy-4-methoxyl substituent and **8** with a 2,4-dihydroxyl substituent showed higher mushroom tyrosinase inhibitory activities than kojic acid, their inhibitory modes of action were investigated using mushroom tyrosinase in the presence of l-dopa. Kinetic studies using Lineweaver–Burk plots were performed using four different concentrations (0, 0.5, 1, and 2 µM) of the test derivatives (**7** and **8**) and five different concentrations (0.5, 1, 2, 4, and 8 mM for **7**, and 1, 2, 4, 8, and 16 mM for **8**) of l-dopa. Lineweaver–Burk double reciprocal plots for derivatives **7** and **8** are shown in Figure 3A,B, respectively. Lineweaver–Burk plots for **7** formed four straight lines that converged on the *y*-axis. The kinetic results of derivative **8** showed a similar pattern to that of derivative **7**. Lineweaver–Burk plots for **8** also formed four straight lines that converged on the *y*-axis. These kinetic results suggest that the BABT derivatives **7** and **8** compete with their natural substrates, namely, l-tyrosine and l-dopa, at the active site of mushroom tyrosinase.

Lineweaver–Burk plots for **7** and **8** were transformed into the corresponding Dixon plots to determine their inhibition constant (K_i_) values for the mushroom tyrosinase–inhibitor complex. Dixon plots for **7** and **8** are shown in Figure 3C,D, respectively. Each Dixon plot formed four different straight lines that converged at one point in the second quadrant of each Dixon plot. The K_i_ values for the mushroom tyrosinase–inhibitor complex were obtained from the *x*-axis value of each convergence point. The K_i_ values were 3.1 × 10^−6^ M for **7** and 8.1 × 10^−7^ M for **8**, indicating that derivative **8** is a stronger mushroom tyrosinase inhibitor than derivative **7**.

### 2.4. In Silico Docking Simulations of BABT Derivatives ***7*** and ***8*** with Various Tyrosinases

#### 2.4.1. In Silico Docking Simulations of BABT Derivatives **7** and **8** with Mushroom Tyrosinase

Because derivatives **7** and **8** showed potent inhibitory effects on mushroom tyrosinase, the binding affinities of these derivatives to the active site of mushroom tyrosinase were predicted using the Schrödinger Suite (release 2021-1) and the mushroom tyrosinase X-ray crystal structure (Protein Data Bank [PDB] ID: 2Y9X). Kojic acid, a well-known tyrosinase inhibitor, was used as a positive control. Tropolone (the original ligand), which binds to the mushroom tyrosinase active site, was deleted, and to the same binding pocket, kojic acid and BABT derivatives **7** and **8** were docked. The binding energies of docked compounds and chemical interactions between compounds **7** and **8** and kojic acid and the amino acid residues present in the active site of mushroom tyrosinase are displayed in 2D and 3D representations in Figure 4.

Three interactions were observed in kojic acid (Figure 4A): a metal coordination between Cu401 and the 2-hydroxymethyl substituent, a π–π stacking interaction between His263 and 4-pyranone, and a hydrogen bond between Met280 and a 5-hydroxyl substituent. Kojic acid achieved a docking score of −4.176 kcal/mol in these interactions. Derivative **7** formed two hydrogen bonds between Ser282 and the 4-methoxyl substituent (as a hydrogen bond acceptor) on the β-phenyl ring of the PUSC template, between Gly281 and the 3-hydroxyl substituent (as a hydrogen bond donor) on the β-phenyl ring, and a π–π stacking interaction between His263 and the phenyl ring of the 2-benzylamino group. These interactions provided **7** with a docking score of −4.179 kcal/mol, which is approximately the same as that of kojic acid. Unlike the binding configuration of derivative **7**, the β-phenyl ring of the PUSC template of derivative **8** was bound close to copper ions. Derivative **8** created two salt bridge interactions with Cu400 and Cu401, using the 4-hydroxyl substituent of the β-phenyl ring, and made three π–π stacking interactions with His259, His263, and Phe264 using the β-phenyl ring of the PUSC template and the phenyl ring of the benzylamino group. These interactions with the amino acid residues of the mushroom tyrosinase active site provided **8** with a docking score of −6.500 kcal/mol, indicating that **8** binds to the active site of mushroom tyrosinase much more strongly than kojic acid. The different arrangements of derivatives **7** and **8** at the mushroom tyrosinase active site were speculated to have a significant effect on binding affinity. These docking results support the kinetic results that derivatives **7** and **8** competitively inhibit the activity of mushroom tyrosinase.

#### 2.4.2. Analysis of Molecular Dynamics Simulation

A molecular dynamics (MD) simulation is typically used to complement and validate the docking results by analyzing the stability and conformational changes of protein–ligand complexes over time [24]. Using Gromacs 2022.2 software (Uppsala University, Uppsala, Sweden), MD simulation was carried out to gauge the flexibility of the mushroom tyrosinase and the overall stability of tyrosinase–ligand complexes. The root mean square deviation (RMSD) and root mean square fluctuation (RMSF) of BABT derivatives **7** and **8** and kojic acid were determined during 100 ns using Gnuplot software.

As shown in Figure 5A, the RMSD of tyrosinase with ligands (**7**, **8**, and kojic acid) was calculated. The RMSD between tyrosinase C-α and ligands was measured. As a result of the measurement, the RMSD between tyrosinase and ligands had a stable value of 2 nm or less, except for some sections. In the case of **7**, it had an overall very low RMSD value of about 0.15 to 0.4 nm. On the other hand, **8** had a higher RMSD value than **7** (0.4 to 0.7 nm), but a lower RMSD value than that of kojic acid. Kojic acid had a similar RMSD value to **7** until 40 ns, but after 40 ns it had an RMSD value approaching 2 nm. RMSD analysis concluded that BABT derivatives **7** and **8** stably bind to tyrosinase, and they show the possibility of showing stronger tyrosinase inhibitory activity than kojic acid.

The RMSF between tyrosinase C-α and ligands is shown in Figure 5B. As a result of the measurement, except for some sections of tyrosinase (300–350 amino acid residues), it was found that the overall RMSF waveforms of **7**, **8**, and kojic acid were similar. This implies that **7** and **8** can have activity similar to that of kojic acid when bound to tyrosinase. In addition, the RMSF values of kojic acid were higher than those of **7** and **8** at tyrosinase residues near positions 125, 150, 200, and 250, indicating that **7** and **8** bind to tyrosinase more strongly than kojic acid. In conclusion, BABT derivatives **7** and **8** are more likely to bind to tyrosinase more tightly than kojic acid and show activity similar to that of kojic acid.

In order to compare the binding strength between tyrosinase and ligands, changes in hydrogen bonding were simulated through MD simulation. Simulation results showed that **7** had more than three hydrogen bonds with tyrosinase at all times, and **8** had fewer hydrogen bonds than **7**, but more than three hydrogen bonds after 90 ns (Figure 5C). Kojic acid was shown to have less than four hydrogen bonds overall and less than two hydrogen bonds in the range of 20–40 ns. In conclusion, MD simulation showed that BABT derivatives **7** and **8** bind to tyrosinase more strongly than kojic acid.

#### 2.4.3. In Silico Docking Simulations of BABT Derivatives **7** and **8** with Human Tyrosinase

Due to the lack of a human tyrosinase X-ray crystal structure, we used a previously constructed human tyrosinase homology model [17,23]. Derivatives **7** and **8** and kojic acid, a positive reference, were docked to the prepared human tyrosinase homology model using the Schrödinger Suite (2021-1) to predict the chemical interactions between each ligand and the amino acid residues of the tyrosinase active site. Unlike derivative **8** and kojic acid, the software did not generate any docking results for derivative **7**. The in silico docking simulation results are shown in Figure 6.

In kojic acid, three types of interactions were observed: a hydrogen bond with Asn364 via the 2-hydroxymethyl substituent, two π–π stacking interactions with His363 and His367 via the 4-pyranone ring, and two salt bridge interactions with Zn6 and Zn7 via the 5-hydroxyl substituent. These interactions provided kojic acid with a docking score of −4.645 kcal/mol. As in the docking results with mushroom tyrosinase, the β-phenyl ring of the PUSC template of derivative **8** was bound close to the metal ions Zn6 and Zn7. Derivative **8** also showed three types of interactions: two salt bridges, a π–π stacking interaction, and two hydrogen bonds. The 4-hydroxyl substituent on the β-phenyl ring generated two salt bridges with Zn6 and Zn7, similar to the 5-hydroxyl substituent on the 4-pyranone ring of kojic acid, and the β-phenyl ring of the PUSC template showed a π–π stacking interaction with His367. The 2-hydroxyl substituent on the β-phenyl ring and amino substituent of the benzylamino group formed hydrogen bonds with Met374 and Glu203, respectively, and these substituents acted as hydrogen bond donors. These interactions with the amino acid residues and Zn metals in the human tyrosinase active site provided **8** with a docking score of −6.841 kcal/mol, which was much lower than that of kojic acid.

### 2.5. Cell Viabilities of BABT Derivatives ***7*** and ***8*** in B16F10 Cells

In addition to exerting potent mushroom tyrosinase inhibitory effects, BABT derivatives **7** and **8** were found to inhibit cellular tyrosinase and melanin production in B16F10 murine melanoma cells. Before these experiments were performed, the influence of these derivatives on cytotoxicity in B16F10 cells was examined to determine their optimal concentration range for B16F10 cell experiments. The cytotoxicity of derivatives **7** and **8** at six concentrations (0, 1, 2, 5, 10, and 20 µM) was assessed using the EZ-Cytox assay.

The viability results measured at three time points (24, 48, and 72 h) are shown in Figure 7. Derivative **7** was not cytotoxic at all concentrations up to 48 h but showed cytotoxicity at a concentration of 20 µM at 72 h. On the other hand, derivative **8** showed no cytotoxicity until 72 h at all tested concentrations. Thus, only derivative **8** was used for B16F10 cell experiments to determine its effects on cellular tyrosinase activity and melanin production.

### 2.6. Effects of BABT Derivative ***8*** on Extracellular and Intracellular Melanin Production in B16F10 Cells

Since BABT derivative **8** had potent mushroom tyrosinase inhibitory activity and no significant cytotoxicity in B16F10 cells at concentrations ≤20 µM, its effect on melanin biosynthesis in B16F10 cells was investigated.

To investigate the effects of derivative **8** on extracellular and intracellular melanin levels in B16F10 cells, the cells were treated with various concentrations (5, 10, and 20 µM) of derivative **8** for 1 h, in addition to co-treatment with α-MSH (0.5 µM) and 3-isobutyl-1-methylxanthine (IBMX; 200 µM) to enhance melanin biosynthesis in B16F10 cells. After 72 h of incubation, the effects of **8** on extracellular and intracellular melanin levels were determined by measuring the melanin absorbance in the culture media and cell lysates, respectively, at 405 nm using a microplate reader. Kojic acid (20 µM) was used as the positive control.

Figure 8A shows the extracellular and intracellular melanin levels. Treatment with stimulators (α-MSH plus IBMX) increased the extracellular melanin levels by 1.7-fold compared to those in the untreated control (100%). Treatment with 20 µM kojic acid significantly reduced the extracellular melanin levels by 1.5-fold, and treatment with derivative **8** significantly and dose-dependently reduced the extracellular melanin levels enhanced by stimulators to 1.22-, 1.03-, and 1.00-fold at 5, 10, and 20 µM, respectively, compared to those in the untreated control. These results implied that **8** at concentrations ≥10 µM had a strong anti-melanogenic effect, returning the melanin levels increased by α-MSH plus IBMX to the normal levels as observed in the untreated control.

Intracellular melanin levels were determined by measuring the absorbance of the cell lysates at 405 nm (Figure 8B). Treatment with stimulators (α-MSH plus IBMX) greatly enhanced the intracellular melanin levels by 2.2-fold compared to those in the untreated control (100%), and kojic acid at 20 µM decreased the intracellular melanin levels enhanced by stimulators to 1.9-fold compared to those in the untreated control. Derivative **8** significantly and dose-dependently reduced the intracellular melanin levels enhanced by stimulators. At 5 µM, **8** showed more potent inhibitory activity on melanin production than kojic acid at 20 µM, and the intracellular melanin levels in cells treated with **8** at 20 µM were lower than those in the untreated control (100%).

### 2.7. Effect of BABT Derivative ***8*** on Tyrosinase Activity in B16F10 Cells

Since derivative **8** was found to exert inhibitory effects on extracellular and intracellular melanin production in B16F10 cells, and tyrosinase acts as a rate-determining enzyme for melanin biosynthesis, the inhibitory effect of **8** on tyrosinase activity was examined using B16F10 cells.

To measure cellular tyrosinase activity in B16F10 cells, cells were treated with various concentrations (5, 10, and 20 µM) of derivative **8** for 1 h, followed by co-treatment with α-MSH (0.5 µM) and IBMX (200 µM) to induce an increase in cellular tyrosinase activity. The cellular tyrosinase inhibitory activity of **8** was assessed by measuring the absorbance at 475 nm (λ_max_ of dopachrome) using a microplate reader. Kojic acid (20 µM) was used as a positive reference control.

Figure 9 shows the cellular tyrosinase activity of **8**. Stimulators (α-MSH plus IBMX) enhanced the cellular tyrosinase activity by 3.2-fold compared to that in the untreated control (100%), and treatment with 20 µM kojic acid reduced the cellular tyrosinase activity by 2.9-fold compared to that in the untreated control. Derivative **8** significantly reduced the cellular tyrosinase activity in a dose-dependent manner. The inhibition of cellular tyrosinase activity by **8** at 5 µM was more than that by kojic acid at 20 µM, and **8** at 20 µM decreased the cellular tyrosinase activity by 1.7-fold compared to that in the untreated control. The inhibitory effect of **8** on cellular tyrosinase activity resembled its inhibitory effect on intracellular melanin production, suggesting that the anti-melanogenic effect of **8** may be mainly due to its inhibitory effect on cellular tyrosinase activity.

### 2.8. Radical Scavenging Effects of BABT Derivatives ***1***–***14*** on 2,2-Diphenyl-1-Picrylhydrazyl (DPPH) and 2,2′-Azino-Bis(3-Ethylbenzothiazoline-6-Sulfonic Acid) (ABTS)

In addition to the direct inhibition of tyrosinase activity, the antioxidant capacity of BABT derivatives may suppress melanogenesis by inhibiting oxidation processes in melanin biosynthesis, starting from tyrosinase substrates (l-tyrosine and l-dopa) [25,26]. To examine other anti-melanogenic mechanisms, the antioxidant activities of BABT derivatives **1**–**14** were evaluated using DPPH and ABTS assays.

DPPH is a light-sensitive compound containing free radicals. The DPPH assay is one of the most common methods for evaluating the antioxidant activity of a compound. The effects of BABT derivatives on DPPH radical scavenging activities were evaluated using l-ascorbic acid as a positive reference control. DPPH radical scavenging activities were measured after leaving the mixed samples (test samples of 500 µM plus DPPH solution of 0.18 mM) in the dark for 30 min. To determine the radical scavenging activities, the absorbance of the mixed samples was measured at 517 nm. Figure 10 shows the DPPH radical scavenging activity of each compound. l-Ascorbic acid exhibited potent DPPH radical scavenging activity with 97% inhibition. Of the 14 BABT derivatives, five BABT derivatives (**5**, **6**, **8**, **9**, and **13**) showed high (76–89% inhibition) and two BABT derivatives (**7** and **12**) showed moderate (47 and 62% inhibition, respectively) DPPH radical scavenging activities. Derivative **9**, with a 3,4-dihydroxyphenyl (catechol) group, showed the most potent DPPH radical scavenging activity with 89% inhibition. Derivative **8**, with a 2,4-dihydroxyphenyl (resorcinol) group, also showed strong DPPH radical scavenging activity with 76% inhibition. Derivatives (**2**–**4**, **10**, **11**, and **14**) with no phenolic hydroxyl substituents showed no significant DPPH radical scavenging activity.

An antioxidant assay using ABTS is also commonly used to assess the antioxidant activities of compounds. Donation of one electron from ABTS to potassium persulfate generates the ABTS radical cation. An ABTS radical scavenging activity assay was performed using 100 µM of BABT derivatives or Trolox (a positive reference control) by mixing test samples with ABTS radical cations obtained from a reaction of potassium persulfate and ABTS. ABTS radical scavenging activity was assessed by measuring the optical density at 734 nm after leaving the test sample–ABTS radical cation mixture in the dark for 2 min. Figure 11 shows the ABTS radical scavenging activities of the test samples. Of the 14 BABT derivatives, **8** showed the most potent ABTS radical scavenging activity with 96% inhibition, similar to that of Trolox (99% inhibition). Derivative **9**, with a catechol substituent, showed potent ABTS radical scavenging activity with 87% inhibition, and four derivatives (**5**, **6**, **7**, and **12**) showed moderate ABTS radical scavenging activities with 43–58% inhibition. As observed in the DPPH radical scavenging assay, derivatives without a hydroxyl substituent on the β-phenyl ring of the PUSC template (**2**–**4**, **10**, **11**, and **14**) did not exhibit any ABTS radical scavenging activity. DPPH and ABTS radical scavenging assay results suggest that derivative **8** may suppress melanogenesis via direct tyrosinase inhibition and indirectly via its antioxidant activities, such as DPPH and ABTS radical scavenging activities.

### 2.9. Scavenging Activities of BABT Derivatives against Reactive Oxygen Species (ROS) and Peroxynitrite

The scavenging activities of the BABT derivatives **1**–**14** against ROS and peroxynitrite (ONOO^−^) were investigated because the antioxidant activities of the compounds may modulate melanogenesis [25,26]. To measure the ROS scavenging activities of BABT derivatives, 2′,7′-dichlorodihydrofluorescein diacetate (DCFH-DA) and 3-morpholinosydnonimine (SIN-1) were used as a fluorogenic probe and ROS generator, respectively. The reaction of DCFH-DA with esterase produces 2′,7′-dichlorodihydrofluorescein (DCFH), which generates the fluorescent compound 2′,7′-dichlorofluorescein (DCF) by reacting with ROS produced by SIN-1. Therefore, ROS scavenging activities were determined by measuring the DCF fluorescence remaining after the BABT derivatives scavenged the ROS generated by SIN-1. Figure 12A shows the ROS scavenging activities of derivatives **1**–**14**. BABT derivatives and Trolox, the positive control, were used at a concentration of 20 µM. SIN-1 treatment significantly increased the ROS levels compared to those in the untreated control, and Trolox reduced the ROS levels enhanced by SIN-1. Three derivatives, namely, **1**, **2**, and **9**, with 4-hydroxyphenyl, 4-methoxyphenyl, and 3,4-dihydroxyphenyl groups, respectively, also strongly reduced the ROS levels enhanced by SIN-1, although the extent of reduction was less than that by Trolox. In addition, four derivatives (**3**, **4**, **5**, and **7**) exhibited moderate ROS scavenging activities. However, derivative **8**, which exerted a potent anti-melanogenic effect in B16F10 cells and mushroom tyrosinase inhibitory effect, did not show significant ROS scavenging effect.

The effects of the BABT derivatives on peroxynitrite scavenging activities were also examined using SIN-1 as an RNS generator and dihydrorhodamine (DHR123) as an ROS indicator. DHR123 was converted to fluorescent rhodamine 123 by peroxynitrite generated by SIN-1. Therefore, the peroxynitrite scavenging activities of **1**–**14** were assessed by measuring the fluorescence of rhodamine 123. BABT derivatives and penicillamine, the positive control, were used at a concentration of 20 µM. Figure 12B shows the peroxynitrite scavenging activities of the BABT derivatives. SIN-1 treatment greatly increased the peroxynitrite levels compared to those in the untreated control, and penicillamine significantly reduced the peroxynitrite levels increased by SIN-1 treatment. Of the 14 BABT derivatives, eight derivatives, including **8**, which showed a potent anti-melanogenic effect in B16F10 cells, scavenged peroxynitrite more potently than did penicillamine. Derivative **1** with a 4-hydroxyphenyl group and derivative **9** with a 3,4-dihydroxyphenyl group showed strong ROS and peroxynitrite scavenging activities.

### 2.10. Effects of BABT Derivative **8** on the Expression Levels of Tyrosinase and Microphthalmia-associated Transcription Factor (MITF) Proteins

We investigated whether, in addition to the direct inhibition of tyrosinase, another inhibitory mechanism of melanogenesis by derivative **8** may be related to the expression of proteins involved in melanogenesis. For the Western blotting experiment, B16F10 cells were treated with α-MSH (1 µM) plus IBMX (200 µM), leading to a 1.7- and 4.9-fold increase in expression levels of melanogenesis-associated proteins tyrosinase and MITF, compared to those in the untreated control (100%) (Figure 13). Treatment with derivative **8** significantly reduced the relative expression levels of tyrosinase and MITF in a dose-dependent manner. Derivative **8** at 10 µM lowered the protein expression to the same or lower control levels. These results suggest that the inhibitory effect of **8** on melanin production may be partially due to its suppression of the expression of melanogenesis-associated proteins.

## 3. Materials and Methods

### 3.1. Chemistry

General Methods

All reagents and solvents were commercially obtained, and anhydrous solvents were prepared using dry methods (distillation over CaH_2_ or Na/benzophenone). All reactions were performed under a nitrogen atmosphere. The progress of the reaction was monitored using thin-layer chromatography (Silica gel 60 F_254_). Flash column chromatography using MP Silica (40–63, 60 Å) was used to purify the reaction mixture. NMR data were recorded using a Varian Unity AS500 unit (Agilent Technologies, Santa Clara, CA, USA) for 125 MHz ^13^C NMR and 500 MHz ^1^H NMR. Chemical shifts (*δ*) were indicated in parts per million (ppm), and coupling constants (*J*) were indicated in hertz (Hz). The splitting patterns were displayed as singlet (s), broad singlet (brs), doublet (d), broad doublet (brd), triplet (t), broad triplet (brt), quartet (q), doublet of doublets (dd), multiplet (m), and broad multiplet (brm).

Only the NMR data of the major tautomers are presented below. ^1^H and ^13^C nuclear magnetic resonance (NMR) spectroscopy and mass (ESI positive and negative modes) spectrometry data of compounds **1**–**16** can be found in Appendix A.

Synthesis of Compound **15 [27]**

A solution of chloroacetyl chloride (7.4 mL, 93.30 mmol) in acetone (20 mL) was added to a solution of benzylamine (5.00 g, 46.66 mmol) in acetone (80 mL) at 0 °C, and the reaction mixture was stirred for 1 h at the same temperature. The volatiles were removed under reduced pressure, and aqueous NaHCO_3_ solution was added to the resultant residue to adjust the pH to 7. The precipitate was filtered and washed with water to obtain **15** (4.50 g, 53%) as a solid.

^1^H NMR (500 MHz, CDCl_3_) *δ* 7.36 (t, 2H, *J* = 7.0 Hz, 3-H, 5-H), 7.33–7.27 (m, 3H, 2-H, 4-H, 6-H), 6.87 (brs, 1H, NH), 4.50 (d, 2H, *J* = 6.0 Hz, benzylic H_2_), 4.10 (s, 2H, CH_2_); ^13^C NMR (125 MHz, CDCl_3_) *δ* 165.9, 137.3, 128.8, 127.8, 127.8, 43.8, 42.6; LRMS (ESI+) *m*/*z* 183.99 (M+H)^+^, calcd 184.05, 206.13 (M+Na)^+^, calcd 206.03, 208 (M+2+Na)^+^, calcd 208.03.

Synthesis of Compound **16** [28]

A solution of **15** (4.43 g, mmol) and ammonium thiocyanate (3.67 g, 48.21 mmol) in ethyl alcohol (100 mL) was refluxed for 7 d. After cooling to ambient temperature, volatiles were evaporated under reduced pressure. The resultant precipitate was filtered using dichloromethane, and the filtrate was evaporated. The resulting residue was purified via silica gel column chromatography using dichloromethane and methanol (30:1) as the eluent to give **16** (2.50 g, 50%) as a solid.

Tautomer ratio = 1:0.2; ^1^H NMR (500 MHz, DMSO-*d*_6_) *δ* 9.65 (brs, 1H, NH), 7.39–7.28 (m, 5H, Ph), 4.62 (s, 2H, benzylic H_2_), 3.94 (s, 2H, CH_2_); ^13^C NMR (125 MHz, DMSO-*d*_6_) *δ* 187.5, 181.3, 138.0, 129.0, 128.0, 127.9, 48.3, 40.5; LRMS (ESI+) *m*/*z* 207.16 (M+H)^+^, calcd 207.06, 229.05 (M+Na)^+^, calcd 229.04.

General Procedure for the Synthesis of BABT Derivatives **1**–**14**

A solution of **16** (100 mg, 0.48 mmol) and appropriate benzaldehyde (1.1 equiv.) in acetic acid (3 mL) was refluxed for 1–7 d in the presence of sodium acetate (119 mg, 1.45 mmol). After adding cold water, the precipitate was filtered and washed with water, dichloromethane, ethyl acetate, and/or methanol, and the filter cake was purified via silica column chromatography using dichloromethane and methanol to obtain **1**–**7** and **9**–**14** as solids with yields of 36–93%. For the synthesis of **8**, **10** was used as the starting material. To a solution of **10** (150 mg, 0.42 mmol) in dichloromethane (2 mL), 1 M BBr_3_ dichloromethane solution (2.6 mL, 2.6 mmol) was added and stirred at room temperature for 3 d. The volatiles were removed under reduced pressure, water was added, and the mixture was filtered. The filter cake was purified via silica gel column chromatography using dichloromethane and methanol (10:1) as the eluent to give **8** (87 mg, 60.5%) as a solid.

Compound **1**

Tautomer ratio = 1:0.2; yield: 93%; ^1^H NMR (500 MHz, DMSO-*d*_6_) *δ* 10.11 (brs, 2H, NH, OH), 7.51 (s, 1H, vinylic H), 7.39 (d, 2H, *J* = 8.0 Hz, 2′-H, 6′-H), 7.36–7.33 (m, 4H, 2-H, 3-H, 5-H, 6-H), 7.29 (t, 1H, *J* = 7.0 Hz, 4-H), 6.88 (d, 2H, *J* = 8.0 Hz, 3′-H, 5′-H), 4.70 (s, 2H, benzylic H_2_); ^13^C NMR (125 MHz, DMSO-*d*_6_) *δ* 180.4, 174.2, 159.5, 137.7, 132.0, 130.4, 129.1, 128.2, 128.0, 125.3, 124.9, 116.7, 48.1; LRMS (ESI−) *m*/*z* 309.19 (M−H)^−^, calcd 309.07, LRMS (ESI+) *m*/*z* 333.24 (M+Na)^+^, calcd 333.07.

Compound **2**

Tautomer ratio = 1:0.2; yield: 55%; ^1^H NMR (500 MHz, DMSO-*d*_6_) *δ* 10.01 (t, 1H, *J* = 5.5 Hz, NH), 7.59 (s, 1H, vinylic H), 7.53 (d, 2H, *J* = 8.5 Hz, 2′-H, 6′-H), 7.40–7.35 (m, 4H, 2-H, 3-H, 5-H, 6-H), 7.31 (t, 1H, *J* = 7.0 Hz, 4-H), 7.08 (d, 2H, *J* = 8.5 Hz, 3′-H, 5′-H), 4.74 (d, 2H, *J* = 5.5 Hz, benzylic H_2_), 3.81 (s, 3H, OCH_3_); ^13^C NMR (125 MHz, DMSO-*d*_6_) *δ* 180.3, 174.1, 160.7, 137.7, 131.7, 129.7, 129.1, 128.2, 128.0, 126.9, 126.2, 115.2, 55.9, 48.1; LRMS (ESI−) *m*/*z* 323.27 (M−H)^−^, calcd 323.03.

Compound **3**

Tautomer ratio = 1:0.1; yield: 49%; ^1^H NMR (500 MHz, DMSO-*d*_6_) *δ* 10.17 (brt, 1H, *J* = 5.5 Hz, NH), 7.58 (s, 1H, vinylic H), 7.57–7.54 (m, 1H), 7.43–7.34 (m, 5H), 7.31–7.26 (m, 2H), 4.74 (d, 2H, *J* = 5.5 Hz, benzylic H_2_); ^13^C NMR (125 MHz, DMSO-*d*_6_) *δ* 179.6, 173.9, 163.2 (dd, *J* = 248.5, 13.3 Hz), 161.1 (dd, *J* = 250.5, 12.3 Hz), 137.5, 131.7, 130.3 (dd, *J* = 10.4, 3.8 Hz), 129.1, 128.3, 128.1, 119.7 (d, *J* = 5.6 Hz), 119.3 (dd, *J* = 11.4, 3.8 Hz), 113.1 (dd, *J* = 21.9, 2.9 Hz), 105.3 (t, *J* = 25.6 Hz), 48.4; LRMS (ESI−) *m*/*z* 329.29 (M−H)^−^, calcd 329.06, LRMS (ESI+) *m*/*z* 353.23 (M+Na)^+^, calcd 353.05.

Compound **4**

Tautomer ratio = 1:0.2; yield: 62%; ^1^H NMR (500 MHz, DMSO-*d*_6_) *δ* 10.14 (s, 1H, NH), 7.64–7.54 (m, 2H), 7.59 (s, 1H, vinylic H), 7.42–7.33 (m, 5H), 7.29 (t, 1H, *J* = 7.0 Hz, 4-H), 4.73 (s, 2H, benzylic H_2_); ^13^C NMR (125 MHz, DMSO-*d*_6_) *δ* 179.7, 173.9, 150.2 (dd, *J* = 249.5, 8.5 Hz), 150.1 (dd, *J* = 244.8, 8.6 Hz), 137.5, 132.4 (dd, *J* = 5.6, 3.8 Hz), 130.5 (d, *J* = 2.9 Hz), 129.1, 128.2, 128.1, 127.8, 127.6, 126.5 (dd, *J* = 6.6, 2.9 Hz), 118.9 (dd, *J* = 18.0, 9.4 Hz), 48.3; LRMS (ESI−) *m*/*z* 329.30 (M−H)^−^, calcd 329.06, LRMS (ESI+) *m*/*z* 353.23 (M+Na)^+^, calcd 353.05.

Compound **5**

Tautomer ratio = 1:0.2; yield: 55%; ^1^H NMR (500 MHz, DMSO-*d*_6_) *δ* 9.95 (s, 1H, NH), 9.72 (s, 1H, OH), 7.54 (s, 1H, vinylic H), 7.41–7.25 (brm, 5H, Ph), 7.14 (s, 1H, 2′-H), 7.02 (brd, 1H, *J* = 8.0 Hz, 6′-H), 6.90 (brd, 1H, *J* = 8.0 Hz, 5′-H), 4.71 (s, 2H, benzylic H_2_) 3.81 (s, 3H, OCH_3_); ^13^C NMR (125 MHz, DMSO-*d*_6_) *δ* 180.3, 174.1, 149.0, 148.3, 137.7, 130.6, 129.0, 128.1, 128.0, 125.8, 125.2, 123.6, 116.5, 114.0, 56.0, 48.1; LRMS (ESI−) *m*/*z* 339.23 (M−H)^−^, calcd 339.08, LRMS (ESI+) *m*/*z* 363.22 (M+Na)^+^, calcd 363.08.

Compound **6**

Tautomer ratio = 1:0.2; yield: 50%; ^1^H NMR (500 MHz, DMSO-*d*_6_) *δ* 9.93 (t, 1H, *J* = 5.5 Hz, NH), 9.65 (s, 1H, OH), 7.52 (s, 1H, vinylic H), 7.38–7.32 (m, 4H, 2-H, 3-H, 5-H, 6-H), 7.29 (t, 1H, *J* = 7.0 Hz, 4-H), 7.12 (d, 1H, *J* = 2.0 Hz, 2′-H), 7.01 (dd, 1H, *J* = 8.5, 2.0 Hz, 6′-H), 6.90 (d, 1H, *J* = 8.5 Hz, 5′-H), 4.71 (d, 2H, *J* = 5.5 Hz, benzylic H_2_), 4.06 (q, 2H, *J* = 7.0 Hz, OCH_2_), 1.34 (t, 3H, *J* = 7.0 Hz, OCH_2_C*H_3_*); ^13^C NMR (125 MHz, DMSO-*d*_6_) *δ* 179.9, 173.6, 148.8, 147.0, 137.3, 130.1, 128.6, 127.7, 127.5, 125.3, 124.7, 123.4, 116.1, 114.6, 63.9, 47.6, 14.7; LRMS (ESI+) *m*/*z* 355.19 (M+H)^+^, calcd 355.11, *m*/*z* 376.92 (M+Na)^+^, calcd 377.09.

Compound **7**

Tautomer ratio = 1:0.1; yield: 55%; ^1^H NMR (500 MHz, DMSO-*d*_6_) *δ* 9.98 (t, 1H, *J* = 5.5 Hz, NH), 9.40 (s, 1H, OH), 7.47 (s, 1H, vinylic H), 7.38–7.33 (m, 4H, 2-H, 3-H, 5-H, 6-H), 7.30 (t, 1H, *J* = 7.0 Hz, 4-H), 7.05–7.00 (m, 2H, 5′-H, 6′-H), 6.99 (s, 1H, 2′-H), 4.72 (d, 2H, *J* = 5.5 Hz, benzylic H_2_), 3.80 (s, 3H, OCH_3_); ^13^C NMR (125 MHz, DMSO-*d*_6_) *δ* 180.3, 174.2, 149.7, 147.3, 137.7, 130.2, 129.1, 128.2, 128.0, 127.1, 126.0, 122.9, 116.0, 112.9, 56.1, 48.1; LRMS (ESI−) *m*/*z* 339.45 (M−H)^−^, calcd 339.08, LRMS (ESI+) *m*/*z* 363.35 (M+Na)^+^, calcd 363.08.

Compound **8**

Tautomer ratio = 1:0.2; 2-step yield: 38%; ^1^H NMR (500 MHz, DMSO-*d*_6_) *δ* 10.20 (brs, 1H), 9.95 (brs, 1H), 9.83 (s, 1H), 7.84 (s, 1H, vinylic H), 7.36 (t, 2H, *J* = 7.0 Hz, 3-H, 5-H), 7.33 (d, 2H, *J* = 7.0 Hz, 2-H, 6-H), 7.29 (t, 1H, *J* = 7.0 Hz, 4-H), 7.15 (d, 1H, *J* = 8.5 Hz, 6′-H), 6.40 (d, 1H, *J* = 2.0 Hz, 3′-H), 6.37 (dd, 1H, *J* = 8.5, 2.0 Hz, 5′-H), 4.69 (s, 2H, benzylic H_2_); ^13^C NMR (125 MHz, DMSO-*d*_6_) *δ* 180.7, 174.3, 161.1, 159.1, 137.9, 129.6, 129.0, 128.1, 128.0, 125.4, 123.2, 112.9, 108.3, 103.0; LRMS (ESI+) *m*/*z* 349.30 (M+H)^+^, calcd 349.06; HRMS (ESI+) *m*/*z* C_17_H_15_N_2_O_3_S (M+H)^+^ calcd 327.0798, obsd 327.0794, *m*/*z* C_17_H_14_N_2_NaO_3_S (M+Na)^+^ calcd 349.0617, obsd 349.0614.

Compound **9**

Tautomer ratio = 1:0.2; yield: 36%; ^1^H NMR (500 MHz, DMSO-*d*_6_) *δ* 9.93 (brt, 1H, *J* = 5.5 Hz, NH), 9.63 (s, 1H, OH), 9.35 (s, 1H, OH), 7.42 (s, 1H, vinylic H), 7.38–7.33 (m, 4H, 2-H, 3-H, 5-H, 6-H), 7.29 (t, 1H, *J* = 7.0 Hz, 4-H), 6.96 (s, 1H, 2′-H), 6.90 (d, 1H, *J* = 8.0 Hz, 6′-H), 6.83 (d, 1H, *J* = 8.0 Hz, 5′-H), 4.71 (d, 2H, *J* = 5.5 Hz, benzylic H_2_); ^13^C NMR (125 MHz, DMSO-*d*_6_) *δ* 180.4, 174.2, 148.1, 146.2, 137.8, 130.7, 129.0, 128.2, 128.0, 125,7, 124.8, 123.3, 116.6, 116.4, 48.1; LRMS (ESI−) *m*/*z* 324.96 (M−H)^−^, calcd 325.06, LRMS (ESI+) *m*/*z* 349.09 (M+Na)^+^, calcd 349.06.

Compound **10**

Tautomer ratio = 1:0.2; yield: 58%; ^1^H NMR (500 MHz, DMSO-*d*_6_) *δ* 9.92 (t, 1H, *J* = 5.5 Hz, NH), 7.80 (s, 1H, vinylic H), 7.40–7.32 (m, 5H, 2-H, 3-H, 5-H, 6-H, 6′-H), 7.29 (t, 1H, *J* = 7.0 Hz, 4-H), 6.68 (dd, 1H, *J* = 8.5, 2.0 Hz, 5′-H), 6.64 (d, 1H, *J* = 2.0 Hz, 3′-H), 4.71 (d, 2H, *J* = 5.5 Hz, benzylic H_2_), 3.86 (s, 3H, OCH_3_), 3.81 (s, 3H, OCH_3_); ^13^C NMR (125 MHz, DMSO-*d*_6_) *δ* 179.9, 173.8, 162.2, 159.4, 137.3, 129.2, 128.6, 127.7, 127.5, 125.6, 123.8, 115.3, 106.1, 98.6, 55.9, 55.6, 47.6; LRMS (ESI−) *m*/*z* 353.30 (M−H)^−^, calcd 353.10, LRMS (ESI+) *m*/*z* 377.25 (M+Na)^+^, calcd 377.09.

Compound **11**

Tautomer ratio = 1:0.2; yield: 52%; ^1^H NMR (500 MHz, DMSO-*d*_6_:CDCl_3_ = 1:0.4) *δ* 9.94 (brs, 1H, NH), 7.55 (s, 1H, vinylic H), 7.36–7.24 (brm, 5H, Ph), 7.12–7.07 (brm, 2H, 2′-H, 6′-H), 7.02 (brd, 1H, *J* = 8.0 Hz, 5′-H), 4.71 (s, 2H, benzylic H_2_), 3.80 (s, 6H, 2×OCH_3_); ^13^C NMR (125 MHz, DMSO-*d*_6_:CDCl_3_ = 1:0.4) *δ* 180.2, 174.1, 150.5, 149.3, 137.5, 130.2, 128.9, 128.1, 127.9, 127.2, 126.3, 123.2, 113.2, 112.3, 56.0, 55.9, 48.2; LRMS (ESI−) *m*/*z* 353.32 (M−H)^−^, calcd 353.10, LRMS (ESI+) *m*/*z* 377.24 (M+Na)^+^, calcd 377.09.

Compound **12**

Tautomer ratio = 1:0.1; yield: 56%; ^1^H NMR (500 MHz, DMSO-*d*_6_) *δ* 9.95 (t, 1H, *J* = 5.5 Hz, NH), 9.11 (s, 1H, OH), 7.54 (s, 1H, vinylic H), 7.38–7.32 (m, 4H, 2-H, 3-H, 5-H, 6-H), 7.29 (t, 1H, *J* = 7.0 Hz, 4-H), 6.87 (s, 2H, 2′-H, 6′-H), 4.71 (d, 2H, *J* = 5.5 Hz, benzylic H_2_), 3.80 (s, 6H, 3′-H, 5′-H); ^13^C NMR (125 MHz, DMSO-*d*_6_) *δ* 180.3, 174.1, 148.6, 138.1, 137.8, 130.8, 129.0, 128.1, 128.0, 125.6, 124.7, 107.8, 56.4, 48.1; LRMS (ESI−) *m*/*z* 369.33 (M−H)^−^, calcd 369.09, LRMS (ESI+) *m*/*z* 393.26 (M+Na)^+^, calcd 393.09.

Compound **13**

Tautomer ratio = 1:0.1; yield: 39%; ^1^H NMR (500 MHz, DMSO-*d*_6_) *δ* 9.93 (t, 1H, *J* = 5.5 Hz, NH), 7.56 (s, 2H, 2′-H, 6′-H), 7.37–7.28 (m, 6H, vinylic H, OH, 2-H, 3-H, 5-H, 6-H), 7.28 (t, 1H, *J* = 7.0 Hz, 4-H), 4.69 (d, 2H, *J* = 5.5 Hz, benzylic H_2_), 1.39 (s, 18H, 6×CH_3_); ^13^C NMR (125 MHz, DMSO-*d*_6_) *δ* 180.4, 174.1, 156.2, 139.8, 137.7, 131.2, 129.0, 128.1, 128.0, 127.2, 125.7, 125.2, 48.1, 35.1, 30.5; LRMS (ESI−) *m*/*z* 421.49 (M−H)^−^, calcd 421.19, LRMS (ESI+) *m*/*z* 445.51 (M+Na)^+^, calcd 445.19.

Compound **14**

Tautomer ratio = 1:0.2; yield: 54%; ^1^H NMR (500 MHz, DMSO-*d*_6_) *δ* 10.04 (brt, 1H, *J* = 5.0 Hz, NH), 7.57 (s, 1H, vinylic H), 7.39–7.27 (m, 5H, Ph), 6.89 (s, 2H, 2′-H, 6′-H), 4.72 (brd, 2H, *J* = 5.0 Hz, benzylic H_2_), 3.81 (s, 6H, 2×OCH_3_), 3.70 (s, 3H, 4′-OCH_3_); ^13^C NMR (125 MHz, DMSO-*d*_6_) *δ* 180.0, 174.1, 153.6, 139.1, 137.6, 130.1, 129.1, 128.2, 128.1, 128.1, 128.0, 107.4, 60.6, 56.4, 48.2; LRMS (ESI−) *m*/*z* 383.32 (M−H)^−^, calcd 383.11, LRMS (ESI+) *m*/*z* 407.26 (M+Na)^+^, calcd 407.10.

### 3.2. In Vitro Assays and In Silico and Kinetic Studies

#### 3.2.1. Mushroom Tyrosinase Inhibition Assay

Mushroom tyrosinase inhibitory activities of BABT derivatives were determined as described previously [15], with minor modifications. l-Tyrosine and l-dopa were used as substrates to evaluate the enzyme activity. In brief, an aqueous solution of mushroom tyrosinase (20 µL, 1000 units/mL) was added to each well of a 96-well microplate containing 170 µL of substrate mixture consisting of 345 μM l-tyrosine, 17.2 mM sodium phosphate buffer (pH 6.5), and 10 µL of BABT derivative (3–6 different concentrations (4, 20, and 100 for **1**–**6** and **9**–**14**; 2, 5, and 10 for kojic acid and **7**; and 0.0625, 0.25, 1, 4, 16, and 64 μM for **8**) in DMSO) or 10 µL of kojic acid. Assay mixtures were incubated at 37 °C for 30 min, and dopachrome amounts synthesized were determined by measuring the optical density of each well at 475 nm using a microplate reader (VersaMax; Molecular Devices, Sunnyvale, CA, USA).

#### 3.2.2. Kinetic Studies of the Efficacies of Derivatives **7** and **8** in Mushroom Tyrosinase Inhibition

Lineweaver–Burk plots of derivatives **7** and **8** were obtained in the presence of l-dopa as the substrate. Briefly, 20 µL aliquots of mushroom tyrosinase aqueous solution (200 units/mL) were added to each well of a 96-well plate containing 10 µL BABT derivative (**7** or **8**) DMSO solution (final concentrations: 0, 0.5, 1, and 2 µM) and 170 µL of an aqueous solution containing l-dopa (final concentrations: 1.0, 2.0, 4.0, 8.0, and 16.0 mM) and sodium phosphate buffer (pH 6.5, 17.2 mM). The initial rate of dopachrome formation was determined by examining the changes in absorbance at a wavelength of 475 nm (ΔOD_475_/min) using a microplate reader (VersaMax; Molecular Devices).

#### 3.2.3. In Silico Study of Chemical Interactions between Kojic Acid and Derivatives **7** and **8** and Mushroom Tyrosinase

An in silico analysis of the chemical interactions between mushroom tyrosinase and the BABT derivatives (**7** and **8**) was performed using the Schrödinger Suite (2021-2), as described previously [17], with minor modifications. The crystal structure of mushroom tyrosinase (*m*TYR) with 2Y9X (*Agaricus bisporus*) PDB ID was imported from PDB to the Protein Preparation Wizard in Maestro 12.4. Unwanted protein chains in the *m*TYR crystal structure were deleted. The structure was optimized by deleting water molecules >3 Å from the enzyme and adding hydrogen atoms. The glide grid of the *m*TYR active site was determined using the binding location of the ligand tropolone, as determined using literature data [29,30,31] and PDB. Kojic acid, a representative tyrosinase, was used as the positive in silico docking reference compound. The structures of in silico docking compounds (kojic acid and derivatives **7** and **8**) were imported to the entry list of Maestro in CDXML format and developed using LigPrep prior to ligand docking. Chemical structures were docked to the tyrosinase glide grid using the Glide task list [32]. Ligand–protein interactions and binding affinities were determined using the glide extra precision (XP) method [33].

#### 3.2.4. Molecular Dynamics Simulation

The Gromacs 2022.2 program was utilized to perform molecular dynamics simulations of the tyrosinase–**7**, **8**, and kojic acid (control) complex structures. Tyrosinase molecular force field parameters were written in Gromos53a6 force field format, and **7**, **8**, or kojic acid molecular force field parameters were derived from Automated Topology Builder (ATB, https://atb.uq.edu.au/index.py, accessed on 30 December, 2022) written in Gromos 54a7 force field format, which was converted into Gromacs format data. At the beginning stage, energy minimization was executed using a steep descent method of 50,000 steps to have a stable conformation. After minimization, canonical ensembles (NVT) and isobar isothermal ensembles (NPT) were performed, respectively, with a constant temperature of 300 K for 100 ps for NVT, followed by a constant temperature of 300 K and a constant pressure of 1 atm per 100 ps for NPT. The production MD runs were then performed for 100 ns, keeping the temperature at 300 K and the pressure at 1 bar. The root mean square deviation (RMSD), root mean square fluctuation (RMSF), and hydrogen bonds between tyrosinase and **7**, **8**, and kojic acid were calculated after the runs. The resulting graphics for these parameters were designed using the Gnuplot program.

#### 3.2.5. In Silico Study of Chemical Interactions between Kojic Acid and Derivative **8** and the Human Tyrosinase Homology Model

A previously created human tyrosinase *h*TYR) homology model was used for a docking simulation with kojic acid and **8** [17,23]. Compounds (kojic acid and **8**) were docked with the *h*TYR model using the protocols described above for in silico *m*TYR docking.

#### 3.2.6. Cell Culture

Murine melanoma cells (B16F10) were purchased from the American Type Culture Collection (Manassas, VA, USA). Streptomycin, Dulbecco’s modified Eagle’s medium (DMEM), penicillin, fetal bovine serum (FBS), trypsin, and phosphate-buffered saline (PBS) were purchased from Gibco/Thermo Fisher Scientific (Carlsbad, CA, USA). B16F10 cells were cultured at 37 °C in DMEM containing streptomycin (100 µg/mL), 10% heat-inactivated FBS, and penicillin (100 IU/mL) in a 5% CO_2_ atmosphere.

#### 3.2.7. Cell Viability Assays

Cytotoxicity assays in B16F10 cells were performed using the EZ-Cytox solution (EZ-3000, DoGenBio, Seoul, Republic of Korea) [13]. Briefly, B16F10 cells were seeded at a density of 1 × 10^4^ cells per well and cultured at 37 °C in a 5% CO_2_ atmosphere for 24 h. The next day, cells were treated with **7** or **8** at six different concentrations (final concentration: 0, 1, 2, 5, 10, and 20 μM) and incubated at 37 °C in a 5% CO_2_ atmosphere for 72 h. Then, the EZ-Cytox solution (10 μL aliquot) was added to each well. After 2 h of incubation at 37 °C, the optical density of each well was measured at 450 nm using a microplate reader (VersaMax; Molecular Devices).

#### 3.2.8. Measurement of Melanin Levels

The effect of **8** on melanin production was measured using a standard melanin content assay [34], with minor modifications. Briefly, B16F10 cells were seeded at a density of 1 × 10^5^ cells per well and allowed to adhere to the well bottoms for 24 h under the conditions used for cell cultures. B16F10 cells were treated with kojic acid (positive reference compound, 20 µM) or four different concentrations (final concentration: 0, 5, 10, and 20 µM) of **8** for 1 h and treated with a stimulator (0.5 µM α-MSH plus 200 µM IBMX). After incubation for 72 h at 37 °C in a 5% CO_2_ atmosphere, melanin levels were determined. To measure the extracellular melanin levels, the absorbance of each well of the culture medium was measured at 405 nm. Intracellular melanin levels were determined as follows: B16F10 cells treated with a stimulator (0.5 µM α-MSH plus 200 µM IBMX) in the presence or absence of test samples (**8** and kojic acid) were cultured for 72 h. Cultured cells were washed twice with PBS, and the pellets obtained were dissolved using 1 N NaOH solution (200 µL, 10% DMSO) at 60 °C for 1 h. Cell lysates were transferred to the wells of a 96-well plate, and the optical density of melanin dissolved in each well was measured at 405 nm in aqueous DMSO solution using a microplate reader (VersaMax; Molecular Devices).

#### 3.2.9. Anti-Tyrosinase Activity Assay

The inhibitory effect of derivative **8** on B16F10 cellular tyrosinase activity was evaluated by measuring the oxidation rate of l-dopa, as previously described [13], with minor modifications. Briefly, B16F10 cells were seeded at 1 × 10^5^ cells/well in a 6-well plate and allowed to adhere to well bottoms for 24 h. B16F10 cells were treated with kojic acid at 20 µM or with **8** at four different concentrations (final concentrations: 0, 5, 10, and 20 µM) for 1 h. Then, a stimulator (0.5 µM α-MSH plus 200 µM IBMX) was applied to increase the cellular tyrosinase activity, and the cells were cultured for 72 h under the conditions used for cell cultures. Cultured cells were washed twice with PBS and exposed to an aliquot (100 µL) of the lysis buffer solution containing 20% Triton X-100 (5 µL), 50 mM phosphate buffer (pH 6.5, 90 µL), and 2 mM phenylmethylsulfonyl fluoride (5 µL) to lyse the cultured cells for 30 min at −80 °C. After defrosting, the cell lysates were transferred to microcentrifuge tubes and centrifuged for 30 min at 4 °C and 12,000 rpm, and the supernatants (80 µL) in each well were mixed with 20 µL l-dopa (2 mg/mL) and incubated at 37 °C for 10 min. The absorbance of dopachrome was measured at 475 nm using a microplate reader (VersaMax; Molecular Devices).

#### 3.2.10. DPPH Radical Scavenging Assay

According to previously described methods [35], with minor modifications, the DPPH radical scavenging activities of BABT derivatives **1**–**14** were measured. A DPPH (0.2 mM) and methanol solution (180 µL) was mixed with 20 µL of a DMSO solution containing 5 mM BABT derivative, or with 20 µL of an aqueous solution of 5 mM l-ascorbic acid, in each well of a 96-well plate and kept at ambient temperature in the dark for 30 min. The optical density of each well was measured at 517 nm using a microplate reader (VersaMax; Molecular Devices). l-Ascorbic acid was used as a positive reference control, and the DPPH radical scavenging abilities of BABT derivatives were compared with those of l-ascorbic acid. All experiments were performed independently, thrice.

#### 3.2.11. ABTS Radical Scavenging Assay

According to a previously described method [36], with minor modifications, the ABTS radical scavenging activities of BABT derivatives **1**–**14** were assessed. Briefly, an ABTS radical solution was prepared by mixing 10 mL of aqueous ABTS solution (7 mM) with 10 mL of potassium persulfate aqueous solution (2.45 mM), and the mixture was kept for 11–15 h in the dark at room temperature until the absorbance remained constant. Prior to the antioxidant assay experiments, the ABTS radical solution was diluted with water to adjust the absorbance at 734 nm to 0.70 ± 0.02. A 10 µL aliquot of BABT derivatives or Trolox at a concentration of 1000 µM was added to 90 µL of ABTS+ free radical solution in each well of a 96-well plate and incubated for 2 min in the dark at ambient temperature. The absorbance of each well was measured at 734 nm using a microplate reader (VersaMax; Molecular Devices). The ABTS+ radical scavenging activities of BABT derivatives were compared to those of Trolox. All experiments were carried out independently, thrice.

#### 3.2.12. SIN-1-Induced ROS Scavenging Assays

The ROS scavenging activities of the BABT derivatives were measured using an oxidant-sensitive fluorescent probe [37,38]. Derivatives or Trolox (10 µL; final concentration: 20 µM) were seeded in a black 96-well plate. Then, 10 µL of SIN-1 (10 µM) was added to induce radical species generation and mixed with 180 µL of sodium phosphate buffer (pH 7.4) containing 12.5 µM DCF-DA and 600 units of esterase. Fluorescence intensity was measured at emission and excitation wavelengths of 535 and 485 nm, respectively, using a fluorescence microplate reader (Berthold Technologies GmbH & Co., Wien, Austria). Trolox was used as the positive control.

#### 3.2.13. Peroxynitrite Scavenging Assays

The peroxynitrite scavenging activities of the BABT derivatives were assessed using the method developed by Kooy [39], involving the monitoring of highly fluorescent rhodamine 123, which was rapidly produced from non-fluorescent DHR123 in the presence of peroxynitrite. Rhodamine buffer (pH 7.4) was composed of 50 mM sodium phosphate dibasic, 50 mM sodium phosphate monobasic, 90 mM sodium chloride, 5 mM potassium chloride, and 100 µM DTPA. A 10 µL aliquot of BABT derivatives or PCA (final concentration: 20 µM) was seeded in a black 96-well plate. Then, 10 µL of SIN-1 (10 µM) was added to induce radical species generation and mixed with 180 µL of rhodamine buffer. Fluorescence intensity was measured at emission and excitation wavelengths of 535 and 485 nm, respectively, using a fluorescence microplate reader (Berthold Technologies GmbH & Co.). PCA was used as the positive control.

#### 3.2.14. Western Blotting Analysis of Tyrosinase and MITF Proteins

B16F10 cells were seeded in a 60 mm dish at a density of 5 × 10^5^ cells/mL for 24 h, washed with cold PBS, and harvested. Cytosolic and nuclear fractions of cell lysates were extracted using lysis buffer, as described previously [13]. Protein concentrations in the cell lysates were estimated using the BCA protein assay (Pierce, Rockford, IL, USA). Equal amounts of protein (20 µg) were loaded and electrophoresed on a 9% sodium dodecyl sulfate–polyacrylamide gel and transferred onto a PVDF membrane (Millipore, Billerica, MA, USA) using a semi-dry system (Bio-Rad, Hercules, CA, USA). The membranes were blocked with a blocking buffer (TBS with 0.05% Tween 20) for 1 h. After blocking, the membranes were incubated with specific primary antibodies at 4 °C overnight. Antibodies against tyrosinase, MITF, β-actin, and transcription factor IIB were purchased from Santa Cruz Biotechnology (Dallas, TX, USA). After incubation, the membranes were washed and incubated for 1 h at room temperature with horseradish peroxidase-conjugated secondary (anti-mouse or anti-goat) antibodies diluted to 1:5000 in 10% TBS buffer. The membranes were washed and protein bands detected using the SuperSignal West Pico Chemiluminescence assay kit (Advansta, San Jose, CA, USA) and the Davinch–Chem program (Davinch-K, Seoul, Republic of Korea). Immunoblotting data were quantified using CS Analyzer 3.2 (Densitograph) image analysis software (http://www.attokorea.co.kr, accessed on 21 November 2022).

#### 3.2.15. Statistical Analysis

One-way analysis of variance, followed by a Bonferroni post hoc test, was used to determine the significant differences between treatments. Statistical analysis was conducted using GraphPad Prism 5 (La Jolla, CA, USA). Results are represented as the mean ± standard error of the mean. Two-sided *p*-values < 0.05 were considered to be significant.

## 4. Conclusions

To identify a new tyrosinase inhibitor, we designed BABT derivatives based on the structure of MHY2081 in this study. Of the 14 BABT derivatives tested, two exhibited strong inhibitory activities against mushroom tyrosinase. BABT derivatives **7** and **8**, which showed stronger mushroom tyrosinase inhibition than kojic acid, were proven to be competitive inhibitors of mushroom tyrosinase in the kinetic study using Lineweaver–Burk plots. In silico docking simulation results using mushroom tyrosinase indicated that they bound to the active site of tyrosinase, supporting the kinetic study results. In experiments using B16F10 cells, **8** exhibited stronger inhibition of melanin production and cellular tyrosinase activity than did kojic acid, and the similar pattern of inhibition of tyrosinase activity and melanin production indicated that the anti-melanogenic effect of **8** was due to tyrosinase inhibition. In addition, **8** potently removed the DPPH and ABTS radicals and peroxynitrite and suppressed melanogenesis-associated protein expression, which indicates its potential as an attractive anti-melanogenic agent. These results suggest that derivative **8** may be used as an effective multifunctional tyrosinase inhibitor.

## Data Availability

Not applicable.

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
