# Peer review of "Design and Synthesis of (Z)-2-(Benzylamino)-5-benzylidenethiazol-4(5H)-one Derivatives as Tyrosinase Inhibitors and Their Anti-Melanogenic and Antioxidant Effects"

_molecules, 2023, doi:10.3390/molecules28020848_

Round 1

Reviewer 1 Report

I have gone through the manuscript submitted by Le and group for the possible publication. The work focuses on the development of BABT derivatives and their utility has been explored as tyrosinase inhibitors and for anti-melanogenic and antioxidant effect. The emerging synthetics are expected to be utilised for their plausible development in the treatment of hyperpigmentation-related diseases. Although the work is interesting i still have some queries that may be duly addressed for a possible publication in the submitted journal by the authors.

The specific queries are as follows: 
1. The drug design portion is not convincing. This section may be improved using the technique of pharmacophore mapping or molecular docking considering the features of MHY2081 or tyrosinase drug binding cavity.
2. In synthetic protocol 'migratory cyclisation' may be defined and duly cited. 3. In the synthetic protocol step two (conversion of 15 to 16) is rate limiting which almost takes 7days for completion. Have authors thought to optimize this? Similarly using acetone as solvent in first step could lead to the side product formation.
4. For mushroom tyrosinase inhibition the value of SEM is relatively very high and therefore it is advisable to repeat the experiment. Also the information on varying concentration of the synthetics employed for inhibitory activities is missing in material and methods section. Do update this.
5. The logP values that are predicted using chemdraw software may be removed. As it is almost same even the substituents are varying from hydrophobic to hydrophilic.
6. In the molecular modelling studies it would be recommended to employ the the technique of molecular dynamics atleast for 100 ns simulation time. The time frame analysis and PL contact plot for both kojic acid and derivative 7 and 8 may be discussed. This will be more convincing approach.
7. Re check all the LRMS values, many of them are incorrect. Author are advised to mentioned the value which they observed and obtained correctly. Also the values must be given atleast upto 2 decimal points.
8. The plots in figure 12 b may be replaced by densitometric plots to improve the readability of the immunoblotting studies.  

Author Response

I have gone through the manuscript submitted by Le and group for the possible publication. The work focuses on the development of BABT derivatives and their utility has been explored as tyrosinase inhibitors and for anti-melanogenic and antioxidant effect. The emerging synthetics are expected to be utilised for their plausible development in the treatment of hyperpigmentation-related diseases. Although the work is interesting i still have some queries that may be duly addressed for a possible publication in the submitted journal by the authors.

The specific queries are as follows: 

  1. The drug design portion is not convincing. This section may be improved using the technique of pharmacophore mapping or molecular docking considering the features of MHY2081 or tyrosinase drug binding cavity.

Thank you for your valuable comments. As suggested by the reviewer, the introduction section was modified to improve the drug design section.

  1. In synthetic protocol 'migratory cyclisation' may be defined and duly cited.

Thank you for your valuable comments. As the reviewer suggested, 'migratory cyclisation' was defined and cited in the synthetic section.

  1. In the synthetic protocol step two (conversion of 15 to 16) is rate limiting which almost takes 7days for completion. Have authors thought to optimize this? Similarly using acetone as solvent in first step could lead to the side product formation.

Thank you for your valuable comments. Step 2 took 7 days to complete the reaction, but yielded sufficient quantities to synthesize all desired final compounds. Therefore, we did not try to optimize the reaction. We appreciate you for the information on step 1 (Using acetone as solvent in first step could lead to the side product formation.).

  1. For mushroom tyrosinase inhibition the value of SEM is relatively very high and therefore it is advisable to repeat the experiment.

Thank you for your kind comments. Mushroom tyrosinase inhibition experiments were repeated with compounds showing high SEM values in the presence of L-tyrosine or L-dopa. According to the results, Table 1 and a related text were revised.

Also the information on varying concentration of the synthetics employed for inhibitory activities is missing in material and methods section. Do update this.

Thank you for your valuable comments. As suggested by the reviewer, we have added information on concentration in Material and Methods section.

  1. The logP values that are predicted using chemdraw software may be removed. As it is almost same even the substituents are varying from hydrophobic to hydrophilic.

Thank you for your valuable comments. As the reviewer suggested, the contents regarding partition coefficient were removed from Table 1 and the text.

  1. In the molecular modelling studies it would be recommended to employ the the technique of molecular dynamics at least for 100 ns simulation time. The time frame analysis and PL contact plot for both kojic acid and derivative 7 and 8 may be discussed. This will be more convincing approach.

Thank you for your valuable suggestions. We performed molecular dynamics (MD) simulation between mushroom tyrosinase and compounds 7, 8, and kojic acid, and data regarding RMSD, RMSF, and the number of hydrogen bonds obtained from MD simulation for 100 ns have been inserted to the manuscript. These results supported results obtained from docking simulation. Please refer to the main text for details.

  1. Re check all the LRMS values, many of them are incorrect. Author are advised to mentioned the value which they observed and obtained correctly. Also the values must be given at least upto 2 decimal points.

Thank you for your meticulous review. We checked all LRMS data and have revised our mistakes. As suggested by the reviewer, we specified the MS values to 2 decimal places, and the observed values were presented without rounding.

  1. The plots in figure 12 b may be replaced by densitometric plots to improve the readability of the immunoblotting studies.  

Thank you for your valuable suggestion. As suggested by the reviewer, we have added densitometric plots to improve the readability of the immunoblotting studies.

Reviewer 2 Report

The Authors of the publication entitled „Design and synthesis of (Z)-2-(benzylamino)-5-benzylidenethiazol-4(5H)-one derivatives as tyrosinase inhibitors and their anti-melanogenic and antioxidant effects” synthesized 14 new benzylidenethiazol-4(5H)-one derivatives and examined their antioxidative activity and as potential tyrosine inhibitors. Based on the research, it can be concluded that they obtained one compound with particularly good activity. In my opinion, the publication is valuable and well written, however, the Authors should provide information on the toxicity of the obtained substances (especially compound 8, did the Authors do such research?) and include an analysis to assess the purity of the obtained substances. This greatly affects the reliability of biological tests (HR MS or elemental analysis or chromatography). The authors should also add the calculated masses to the description of the MS analysis.

Author Response

The Authors of the publication entitled „Design and synthesis of (Z)-2-(benzylamino)-5-benzylidenethiazol-4(5H)-one derivatives as tyrosinase inhibitors and their anti-melanogenic and antioxidant effects” synthesized 14 new benzylidenethiazol-4(5H)-one derivatives and examined their antioxidative activity and as potential tyrosine inhibitors. Based on the research, it can be concluded that they obtained one compound with particularly good activity. In my opinion, the publication is valuable and well written, however, the Authors should provide information on the toxicity of the obtained substances (especially compound 8, did the Authors do such research?) and include an analysis to assess the purity of the obtained substances. This greatly affects the reliability of biological tests (HR MS or elemental analysis or chromatography). The authors should also add the calculated masses to the description of the MS analysis.

Thank you for your valuable comments. As the reviewer suggested, we added HR mass data of compound 8 showing the most potent tyrosinase inhibitory activity and anti-melanogenic effect, and inserted the calculated masses of each compound to the description of the MS analysis. Before cell experiments, we performed a cytotoxicity assay of each compound using an EZ-Cytox assay. Compounds showing cytotoxicity was excluded from cell experiments.

For the purity assessment of compound 8 showing potent tyrosinase inhibitory activity and anti-melanogenic effect, the HR-MS analysis was performed, and the results have been added to the Materials and Methods section and Supplementary information, respectively.

In addition, as suggested by the reviewer, the calculated masses were added to the description of the MS analysis section.

Reviewer 3 Report

Authors in this manuscript synthesized and demonstrated the biological activity of 14 new tyrosinase inhibitors based on the structure of previously synthesized inhibitor, MHY2081. The work is presented with acceptable novelty and good results. Therefore, I would recommend accepting this manuscripts after making the following corrections:

1. In line 27, "via silico docking" should be written via in silico docking.

2. Figure 1 should be referred to in the introduction.

3. The IC50 of MHY2081 should be written in figure 1, so the reader can compare the activity between the parent tyrosinase inhibitor and the newly synthesized derivatives.

4. On the third arrow in scheme 1, 14 benzaldehydes instead of 13.

5. In scheme 1, All R groups should be written. 

6. In section 2.2. it was mentioned that the IC50 values of the synthesized derivatives were obtained from the inhibition tests performed at 3-6 different concentrations. Could you explain why some derivatives' IC50 values were determined with less than 5 concentrations.

7. Spacing between line 224 and 225 should be removed.

8. In section 2.3. it was mentioned that compound 7 has higher tyrosinase inhibitory activity than Kojic acid however, the difference is less than two fold and no statistical analysis was performed, So I suggest performing statistical analysis to make it clear whether there is a significant difference or not or instead you can write the compound 7 has a comparable result with KA.

Author Response

Authors in this manuscript synthesized and demonstrated the biological activity of 14 new tyrosinase inhibitors based on the structure of previously synthesized inhibitor, MHY2081. The work is presented with acceptable novelty and good results. Therefore, I would recommend accepting this manuscripts after making the following corrections:

  1. In line 27, "via silico docking" should be written via in silico docking.

Thank you for your valuable comments. We have revised it.

  1. Figure 1 should be referred to in the introduction.

Thank you for your valuable comments. We have referred Figure 1 in the Introduction.

  1. The IC50 of MHY2081 should be written in figure 1, so the reader can compare the activity between the parent tyrosinase inhibitor and the newly synthesized derivatives.

Thank you for your valuable comments. We have added the IC50 value of MHY2081 in Figure 1.

  1. On the third arrow in scheme 1, 14 benzaldehydes instead of 13.

Thank you for your valuable comments. 13 benzaldehydes are correct because final compound 8 with a 2,4-dihydroxyphenyl group was prepared from final compound 10 with a 2,4-dimethoxyphenyl group via demethylation.

  1. In scheme 1, All R groups should be written. 

Thank you for your valuable comments. As suggested by the reviewer, all R groups were indicated in Scheme 1.

  1. In section 2.2. it was mentioned that the IC50 values of the synthesized derivatives were obtained from the inhibition tests performed at 3-6 different concentrations. Could you explain why some derivatives' IC50 values were determined with less than 5 concentrations.

Thank you for your valuable comments. First, three different concentrations were used for inhibition testing to obtain approximate IC50 values. We then determined more accurate IC50 values using six selected concentrations based on the approximate IC50 values.

  1. Spacing between line 224 and 225 should be removed.

Thank you for your meticulous review. We removed the spacing, as suggested by the reviewer.

  1. In section 2.3. it was mentioned that compound 7 has higher tyrosinase inhibitory activity than Kojic acid however, the difference is less than two fold and no statistical analysis was performed, So I suggest performing statistical analysis to make it clear whether there is a significant difference or not or instead you can write the compound 7 has a comparable result with KA.

Thank you for your meticulous review. We performed statistical analysis to make it clear whether there is a significant difference or not. In the presence of L-tyrosine, there was a significant difference between compound 7 and kojic acid (p > 0.01), while there was no difference in the presence of L-dopa. Because there is a significant difference in the presence of L-tyrosine, use of the text (compound 7 has higher tyrosinase inhibitory activity than kojic acid) appears acceptable. Please kindly understand.

Reviewer 4 Report

At first reading, the manuscript seems very interesting, complex with a solid scientific basis.  However, upon a closer look, there are certain aspects that raise serious question marks.

It is stated that a homology model was performed (line 89, sections 4.2.3. and 4.2.4.), but from the description it appears that you used the same homology model already published in reference 17 (doi: 10.3390/antiox11050948), sections 2.2.4. and 2.2.5. The description of the homology model building is the same as that presented in reference 17. The same methodology, identical protein sequence (hTYR (P14679)) and same template (5M8Q PDB ID).

No specific data was provided with reference to the homology model, sequential alignments or evidence of its validity such as Ramachandran plot, evaluation with Procheck or MolProbity server.

Unfortunately, even if this could have been an interesting study, it is compromised due to the aforementioned aspects.

Round 2

Reviewer 1 Report

The authors have incorporated the points raised. The paper is acceptable in its current form.

Author Response

The authors have incorporated the points raised. The paper is acceptable in its current form.

Thank you for your valuable comments and suggestions.

Reviewer 4 Report

Considering the changes made in the text, I agree that this manuscript could be published after a minor revision. The sentence "In addition, in silico docking simulations with mushroom tyrosinase and the human tyrosinase homology model were performed using the Schrodinger Suite." (lines90-92) must be modified so that it is understood that this homology model was previously made.

Author Response

Considering the changes made in the text, I agree that this manuscript could be published after a minor revision. The sentence "In addition, in silico docking simulations with mushroom tyrosinase and the human tyrosinase homology model were performed using the Schrodinger Suite." (lines90-92) must be modified so that it is understood that this homology model was previously made.

Thank you for your detailed review. As suggested by the reviewer, the sentence have been modified. “In addition, in silico docking simulations with mushroom tyrosinase and the previously prepared human tyrosinase homology model were performed using the Schrodinger Suite.”